# Non-Contrast MR Lymphography and Intranodal Dynamic Contrast MR Lymphangiography in Children with Congenital Heart Disease—Imaging Findings as well as Impact on Patient Management and Outcome

**DOI:** 10.3390/ijms241914827

**Published:** 2023-10-02

**Authors:** Christoph Bauer, Mario Scala, Pavel Sekyra, Franz Fellner, Gerald Tulzer

**Affiliations:** 1Department of Paediatric Cardiology, Kepler University Hospital GmbH, Krankenhausstrasse 26–30, 4020 Linz, Austriagerald.tulzer@kepleruniklinikum.at (G.T.); 2Johannes Kepler University Linz, Altenbergerstrasse 68, 4040 Linz, Austria; pavel.sekyra@kepleruniklinikum.at (P.S.); franz.fellner@kepleruniklinikum.at (F.F.); 3Central Radiology Institute, Kepler University Hospital GmbH, Krankenhausstrasse 9, 4020 Linz, Austria

**Keywords:** lymphatic imaging, protein-losing enteropathy, plastic bronchitis, postoperative chylothorax, Noonan syndrome, lymphatic flow disorders

## Abstract

Lymphatic flow disorders are rare but devastating complications in children with congenital heart disease. T2-weighted magnetic resonance lymphography and intranodal dynamic contrast magnetic resonance lymphangiography are imaging modalities that can depict central lymphatic anatomy and flow pattern. Our objective was to describe the technical aspects and our imaging findings of central lymphatic abnormalities and their impact on patient management and outcomes: We conducted a retrospective review of 26 children with congenital heart disease who presented for lymphatic imaging between 2015 and 2020 at our institution. Eleven had postoperative chylothorax, six had plastic bronchitis, seven had protein-losing enteropathy and three had Noonan syndrome. Our lymphatic imaging demonstrated severely abnormal lymphatic flow in all of the children, but only minor abnormalities in protein-losing enteropathy. No major procedure-related complication occurred. Lymphatic interventions were performed in six patients, thoracic duct decompression in two patients and chylothorax revision in three patients. This led to symptomatic improvements in all of the patients: Lymphatic imaging is safe and essential for the diagnosis of lymphatic flow disorders and therapy planning. Our intranodal lymphangiography depicts an abnormal lymphatic flow pattern from the central lymphatics but failed to demonstrate an abnormal lymphatic flow in protein-losing enteropathy. These imaging techniques are the basis for selective lymphatic interventions, which are promising to treat lymphatic flow disorders.

## 1. Introduction

Lymphatic flow disorders in children with congenital heart disease are rare but devastating complications. Postoperative chylothorax affects over 2.8% of pediatric patients after cardiac surgery, and the long-term risk for the development of protein-losing enteropathy or plastic bronchitis in single ventricle congenital heart disease is 5–15%. Treatment strategies are limited, and mortality and morbidity are still high, especially in Fontan patients [1].

Non-contrast T2-weighted MR lymphography and dynamic contrast MR lymphangiography are new imaging modalities that allow for the visualization of central lymphatic anatomy, lymphatic fluid distribution and lymphatic flow pattern.

Based on these techniques, new treatment strategies have been developed recently with promising results. Percutaneous embolization of lymphatic fistulas has been shown to improve symptoms in 88% of patients with plastic bronchitis, and there are an increasing number of reports of patients who underwent thoracic duct decompression leading to the resolution of protein-losing enteropathy [2,3,4,5,6,7,8]. In patients suffering from postoperative chylothorax, lymphatic embolization has been shown to resolve chylothorax in two distinct etiologies: pulmonary lymphatic perfusion syndrome and traumatic leak, but not in central lymphatic flow disorder [9].

Regarding the growing number of surviving children with congenital heart disease, more patients will need reliable investigations that serve as basis for the proper management of lymphatic disorders [10].

The aim of this study was to analyze and report our institutional experience with non-contrast T2-weighted MR lymphography and intranodal dynamic contrast MR lymphangiography as well as their impact on patient management and outcome in children with congenital heart disease.

## 2. Results

Table 1 summarizes the characteristics and clinical information of all 26 patients. MR lymphography depicted abnormal lymphatics in all of the children. Dynamic contrast MR lymphangiography was technically successful in all but one patient, for whom the procedure’s failure was due to needle dislocation. Lymphatic imaging was performed without general anesthesia in two patients. Procedural-related complications occurred in one child, who developed a minor local infection at the puncture site. During anesthesia, one patient experienced a drop in saturation, and another patient suffered from severe hypoxemia that led to a need for extracorporeal membrane oxygenation support. Both of these children had severe plastic bronchitis.

Postoperative Chylothorax: The latest surgical repair, imaging findings, etiology, resulting interventions and outcome data of all 11 patients with postoperative chylothorax are listed in Table 1 and Table 2.

Lymphatic imaging was able to determine the etiology of the chylothorax in 9 of the 11 patients. Only one of these patients had a traumatic leak (9%), and this was the only patient who did not show severe lymphatic abnormalities (type 3 and 4) in the MR lymphography images. All four children with central lymphatic flow disorders (36%) were younger than 1 year (a median of 93 days; a range of 22–270 days), all had concurrent ascites and three had concurrent body wall edema. The four patients with pulmonary lymphatic perfusion syndrome (36%) were older (a median of 31 months; a range of 9–45 months).

Chylothorax revision was performed in three patients (numbers 4, 8 and 11), and their chylothoraces resolved after 8, 19 and 20 days. Glenn takedown was performed in another infant (number 12), ultimately leading to the resolution of the chylothorax 37 days after the procedure. One infant with central lymphatic flow disorder died. In the remaining patients, their chylothoraces resolved spontaneously.

Plastic bronchitis: Six patients presented with symptoms of plastic bronchitis. One had concurrent protein-losing enteropathy. Table 3 summarizes the details of our MR lymphography and dynamic contrast MR lymphangiography findings, management and outcomes.

The MR lymphography and dynamic contrast MR lymphangiography both demonstrated severely abnormal lymphatics and abnormal lymphatic flow in all of the patients. The dynamic contrast MR lymphangiography confirmed retrograde lymphatic flow from the thoracic duct toward the lung parenchyma or the lymphatic perfusion of the mediastinum in all but one patient whose dynamic contrast MR lymphangiography was technically limited. However, distinct fistulas to the lung could not be identified clearly in two of the patients. Based on these findings, selective lymphatic interventions were successfully performed in three children, and all of them had improvements in symptoms.

The patient with concurrent protein-losing enteropathy received thoracic duct decompression as previously described, resulting in normalized hypoalbuminemia and no more plastic bronchitis symptoms.

Protein-losing enteropathy: In all six children with protein-losing enteropathy (Table 4) whose thoracic duct was intact and unobstructed, the MR lymphography depicted only mild lymphatic abnormalities (Types 1–2).

The dynamic contrast MR lymphangiography was not able to depict fistulas in any of these children. One patient had lymphangiography via liver lymphatics in addition to dynamic contrast MR lymphangiography. A fistula was seen there so a lymphatic intervention was performed. The hypalbuminemia improved after the intervention. Another patient received thoracic duct decompression due to his poor clinical condition, and his hypalbuminemia finally improved. However, he is currently still scheduled for cardiac transplantation.

Noonan Syndrome: The details of our MR lymphography and dynamic contrast MR lymphangiography findings as well as the preprocedural management and outcome of the three patients with genetically confirmed Noonan syndrome are listed in Table 5. Figure 1a,b represent the typical findings of the Noonan syndrome patients. All three children’s lymphatics as well as lymphatic flow were severely abnormal. Lymphatic fistulas to the lung could be visualized in two children.

One of these two patients (number 1) underwent glue embolization of the left and partially the right thoracic duct which were seen during the lymphangiography via the thoracic duct, and their symptoms improved temporarily (at the 10-month follow up). The pleural effusion resolved spontaneously after atrial septal defect closure in the patient (number 2) who did not have fistulas.

## 3. Discussion

Lymphatic imaging procedure and imaging findings: The MR lymphography was reliable in demonstrating abnormal lymphatic anatomy and allowed for the classification of lymphatic abnormality types from the beginning of our program in all of our patients. However, our imaging protocols improved substantially over time. The dynamic contrast MR lymphangiography was technically more challenging at the beginning with a learning curve in how to gain and maintain lymphatic access via the lymph nodes as they were sometimes not even one millimeter in diameter. The key was having an experienced team that could act in sync and a strategy to secure needle positioning and limit motion during patient transfer into the MR. General anesthesia is advisable even in very compliant teens as the whole procedure is long and small movements can lead to needle dislocation.

When we started the lymphatic program, we first learned the basic skills from an experienced center in the USA and adopted the procedures to our local preconditions. The challenges were the differences in workstations and availability of medical equipment. Importantly, we started to review and discuss the patients and MRIs together with experienced colleagues from a leading center in the USA. Later, we constantly reevaluated and refined the procedures ourselves. Another important step in bringing our program forward was to build up a specialized multidisciplinary team. Most of the patients are multimorbid and difficult to manage. Therefore, it is important to have anesthesiologists and gastroenterologists, nurses and intensive care staff who can deal with the unique demands of this special patient population. At least it was an improvement to be able to connect with other people who are doing lymphatic imaging and interventions worldwide and discuss cases and exchange experiences. The improvement at our center clearly not only led to better image quality but also to a better visualization of underlying lymphatic abnormalities and concurrently to better therapy planning.

Dynamic contrast MR lymphangiography is a safe procedure. We only had one minor complication and that was a local infection at the puncture site which was successfully treated with antibiotics. However, we used a semi-sterile approach to avoid infections. Anesthetic complications were more relevant in our children. In particular, patients with plastic bronchitis are at risk of hypoxemia during anesthesia. To reduce this risk, careful patient preparation and proper peri- and postprocedural management are necessary.

However, there are several other severe potential complications that can occur during conventional fluoroscopic lymphangiography and catheter-based lymphatic interventions. During these procedures, Lipiodol is often used to visualize central lymphatic structures as it is more sufficient than the iodinated contrast medium. Lipiodol is an agent that can cause embolisms and strokes. Other important complications of lymphatic interventional procedures are SIRS, pain, bleeding, and pancreatitis. In addition, when glue is administered into lymphatic fistulas, there is always the possibility that it will reflux into functional lymphatic compartments and promote downstream lymphatic dysfunction. To avoid theses complications, all patients should be screened for potential right-to-left shunts. All veno-venous collateral vessels and lympho-pulmonary vein connections are closed before an intervention. Sometimes an open fenestration in a Fontan patient may be temporary closed during the procedure but blocking a fenestration with a balloon may pose a risk of thrombus formation and a subsequent embolism, especially during an intervention in which anticoagulation is prohibited.

Impact on patient management: In the majority of our patients with postoperative chylothoraces, dynamic contrast MR lymphangiography was able to be used to determine an etiology, most commonly pulmonary lymphatic perfusion syndrome and central lymphatic flow disorder. The proper identification of etiology has been shown to be important for the accurate selection of a therapeutic strategy in a small study of 25 patients with congenital heart disease.

In this study, lymphatic embolization was successful in patients with pulmonary lymphatic perfusion syndrome and traumatic leak but not in those with central lymphatic flow disorder. All of the nine described patients with central lymphatic flow disorder ultimately died despite interventions, and all of the patients with pulmonary lymphatic perfusion syndrome and traumatic leaks survived [9].

Lymphatic embolization had not been performed in our patients. Dynamic contrast MR lymphangiography was able to be used to demonstrate retrograde lymphatic flow, but lymphatic fistulas were detected in only two patients. Nevertheless, lymphatic imaging was important to guide surgical chylothorax revision in this patient group as surgeons could focus on the retrograde flow pattern.

Dynamic contrast MR lymphangiography was effective in depicting fistulas originating from the thoracic duct to the lung in patients with plastic bronchitis. The typical picture in this patient group was of retrograde lymphatic flow from the thoracic duct to the lungs. However, we had one patient in which additional fistulas could be visualized during cardiac catheterization in which Lipiodol was directly injected into their thoracic duct. This may increase the contrast concentration/centripetal lymphatic flow and help to depict fistulas not seen on dynamic contrast MR lymphangiography; therefore, it should be considered for a workup in this patient group.

Intranodal dynamic contrast MR lymphangiography may be limited by the amount of contrast medium getting inside the thoracic duct—this could be optimized using a double- or triple-dose contrast agent.

On the other hand, patients with protein-losing enteropathy not only displayed minor lymphatic abnormalities on MR lymphography but also presented no retrograde lymphatic flow and no fistulas on intranodal dynamic contrast MR lymphangiography. In one of these patients, conventional lymphangiography was performed via the liver lymphatics afterwards and revealed a fistula from the liver to the duodenum. This investigation goes along with recently published data from Dori et al. describing two new lymphatic imaging methods that depict mesenteric and liver lymphatics more accurately. In this study, intranodal dynamic contrast MR lymphangiography was also not able to depict duodenal leaks in eleven PLE patients, but intrahepatic and intramesenteric dynamic contrast MR lymphangiography could visualize them in nine and seven of these eleven PLE patients, respectively [11]. The typical flow pattern in PLE patients is therefore hepatoduodenal connections and less often flow from the mesenteries to the duodenum.

The reason for these investigations may reflect the involvement of different lymphatic structures in disease development in plastic bronchitis and protein-losing enteropathy despite having a similar etiology [11,12]. In plastic bronchitis, fistulas usually originate from central lymphatic structures that are well depicted in dynamic contrast MR lymphangiography. In protein-losing enteropathy patients, liver and mesenteric lymphatics are involved and hence are not detected by intranodal dynamic contrast MR lymphangiography. Interventional therapeutic approaches are different in PLE and plastic bronchitis/chylothorax patients; some patients have multicompartment lymphatic failure for which the treatment of one compartment could worsen lymphatic insufficiency in another compartment. So, before the initiation of any treatment, it is necessary to get a whole picture of the abnormal lymphatics.

Particularly in PLE patients, the implementation of additional imaging modalities, most notably intrahepatic and intramesenteric dynamic contrast MR lymphangiography, is useful [13]. This has been supported by a recent single-center retrospective study of 41 consecutive patients in which the authors could show that patients with PLE were more likely to have duodenal involvement on IH-DCMRL than patients without PLE. [14]

For Noonan syndrome, our investigations are in line with previously published data that recently characterized lymphatic abnormalities in this cohort by retrograde intercostal flow, pulmonary lymphatic perfusion and thoracic duct abnormalities [15]. Besides the characterization of lymphatic abnormalities, the confirmation of lymphatic fistulas enabled us to conduct a selective lymphatic intervention in one patient.

Outcome: In our cohort, all the children with pulmonary lymphatic perfusion syndrome and traumatic leaks survived, and their chylothoraces resolved either spontaneously or after surgery. In contrast to the study published by Dori [9], the majority of our children with central lymphatic flow disorder survived. The differences in mortality may be explained by the different population referred to our hospital, assuming that the cohort Dori described was sicker.

Regarding the high number of spontaneous resolutions of chylothorax and the delay from operation to resolution, it is not clear whether surgical revision had any effect in our patients despite guidance.

Larger prospective studies are needed to determine the impact of surgical and interventional therapies on chylothorax resolution after heart operations in these children.

However, the selective glue embolization of thoracic duct branches or fistulas led to symptomatic improvements in all of our patients with plastic bronchitis, protein-losing enteropathy and Noonan syndrome when it was performed. Plastic bronchitis and protein-losing enteropathy symptoms were constantly present in our patients before the intervention. So, a causative effect of the interventions is likely, although we do not have a control group.

Our investigations are therefore in line with previously published data and support the efficacy of these new treatment strategies. Still, the follow-up period of this and other cohorts is short, and their long-term efficacy still has to be determined in further studies.

Thoracic duct decompression is a new and effective therapeutic strategy that was used in two patients with symptoms of protein-losing enteropathy and led to improvements in the protein levels of both of them. In contrast to the embolization of lymphatics, it effectively reduces the pathophysiologic burden and hence may lead to a sustainable improvement.

## 4. Materials and Methods

We retrospectively reviewed the data from 26 children with congenital heart disease (age range from 3 weeks to 17 years; 13 females, 13 males), who presented at our institution for lymphatic imaging because of suspected lymphatic disorders between July 2015 and June 2020. Eleven had postoperative chylothorax (42%), five plastic bronchitis (19%), six protein-losing enteropathy (23%), one protein-losing enteropathy and plastic bronchitis (4%) and three Noonan syndrome (12%). Patient demographics, cardiac diagnoses, surgical history, and the clinical course were gathered from the electronic hospital information system.

MR images were analyzed by two experienced pediatric radiologists with a pediatric cardiologist and discussed until a consensus was reached. The study was approved by the ethics committee of our university prior to its initiation.

### 4.1. Lymphatic Imaging Procedure

All procedures but two were performed under general anesthesia on a 3 Tesla scanner (Magnetom Skyra, Siemens Healthineers, Erlangen, Germany). At first, inguinal lymph nodes were accessed outside the MR site on both sides under semi-sterile conditions using an ultrasound for guidance and a 25-gauge spinal needle. After confirming needle position by injecting 1 mL of 0.9% saline solution, the patient was carefully transferred into the MR site.

Body and spine coils were applied for a large coverage as well as optimal signal-to-noise ratio.

T2-weighted MR lymphography was the first step. It consisted of a respiratory navigated and cardiac-gated high-resolution highly T2-weighted 3-dimensional sequence in coronal plane. The following imaging parameters are meant as an orientation and depend on field of view, resolution, slice number and others.

Typical parameters were a flip angle mode constant with fixed angle around 120° with fat suppression, TE 600–800 and TR 1200–1800 (depending on respiration cycle time).

For dynamic contrast MR lymphangiography, a weight-adjusted amount (0.2 mL/kg) of a macrocyclic gadolinium contrast agent was injected into each inguinal lymph node within one to two minutes. Scanning was initiated immediately after administration using short (20–25 s) 3-dimensional T1-weighted high-resolution sequences in coronal plane and repeated at regular intervals until contrast medium reached left venous angle of the neck. We usually use sequences with the following parameters: shortest possible TE (about 1 ms) and TR (about 3 ms) and flip angle of about 20°. Between these T1 sequences, there are a few short (20–25 s) 3-dimensional T1-weighted volume examination sequences in breath hold with a better anatomical resolution. Typical parameters for this sequence are a flip angle about 12° and minimal TR and TE, such as flash. Both T1-weighted sequences are targeted to get the best possible picture quality in the shortest period of time with an acceptable signal-to-noise ratio. At the end, maximum intensity projections of the contrast agent flow in the lymphatic duct are processed [16].

### 4.2. Image Analysis

Increased signal of lymphatic structures on T2-weighted imaging was considered as abnormal lymphatics. Abnormal lymphatic flow was defined as contrast agent passing from the injection site to any anatomical structure not typically involved in centripetal lymphatic flow to the left venous angle, as previously described by David Biko. Based on the same publication, we described dermal backflow (Figure 1a) and intercostal lymphatic flow (Figure 1b) when intranodal contrast passed retrograde into the intercostal space along the chest wall or into the subcutaneous tissue [13].

Lymphatic fistulas (Figure 2) were described when there was a distinct structure with retrograde flow depicted on dynamic contrast MR lymphangiography.

Otherwise, we described lymphatic leak (Figure 3) when there was a sole contrast accumulation.

Postoperative chylothorax was classified as previously introduced by Dori et al. based on dynamic contrast MR lymphangiography and intranodal lymphangiography and are described as traumatic leak (Figure 3), pulmonary lymphatic perfusion syndrome (abnormal pulmonary lymphatic flow from the thoracic duct toward the lung parenchyma through abnormal lymphatic networks in the chest and/or lymphatic perfusion of the mediastinum) (Figure 4) or central lymphatic flow disorders (abnormal central lymphatic flow, effusion in more than one compartment and the presence of dermal backflow) (Figure 5a,b) [9].

In all patients, lymphatic abnormalities were scored on T2-weighted MR according to Dori et al. Type one was scored when there was unilateral supraclavicular increased signal intensity, type two when there was bilateral increased supraclavicular signal intensity, type three when there was an additional extension into the mediastinum and type four when there was an increased pattern in the lungs [14].

Figure 6 represents typical findings of patients with plastic bronchitis.

### 4.3. Statistical Methods

Demographic and procedural patient variables were summarized using standard descriptive statistics.

## 5. Conclusions

Non-contrast MR lymphography and dynamic contrast MR lymphangiography are safe and reliable lymphatic imaging methods that are essential for the diagnosis of lymphatic flow disorders and therapy planning in patients with congenital heart disease. Intranodal dynamic contrast MR lymphangiography depicts abnormal lymphatic flow patterns that originate from central lymphatic structures in most patients except those with protein-losing enteropathy.

MR lymphography and dynamic contrast MR lymphangiography are important for targeted lymphatic interventions that are new promising treatment approaches to improve symptoms in certain patients.

## Figures and Tables

**Figure 1 ijms-24-14827-f001:**
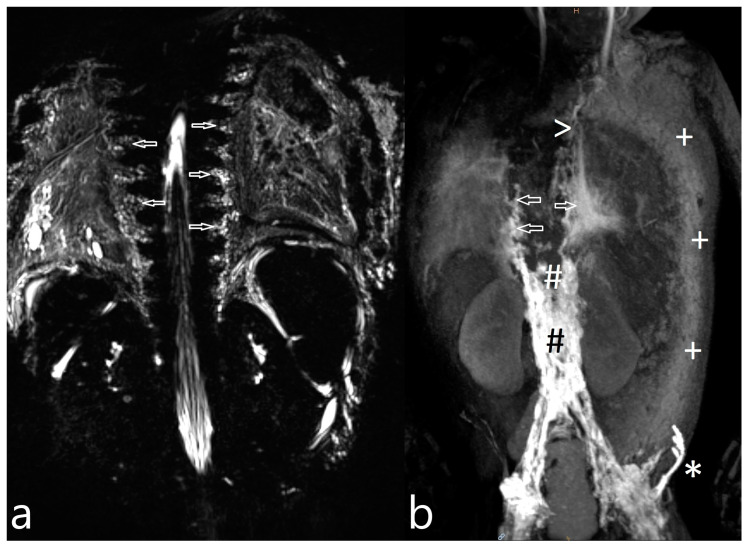
Intercostal lymphatic flow (**a**, arrows point at it) and dermal backflow (**b**, * shows dermal backflow, # is the dilated lymphatic network, arrows point to a doubled thoracic duct with pulmonary effusions, + shows dermal effusions).

**Figure 2 ijms-24-14827-f002:**
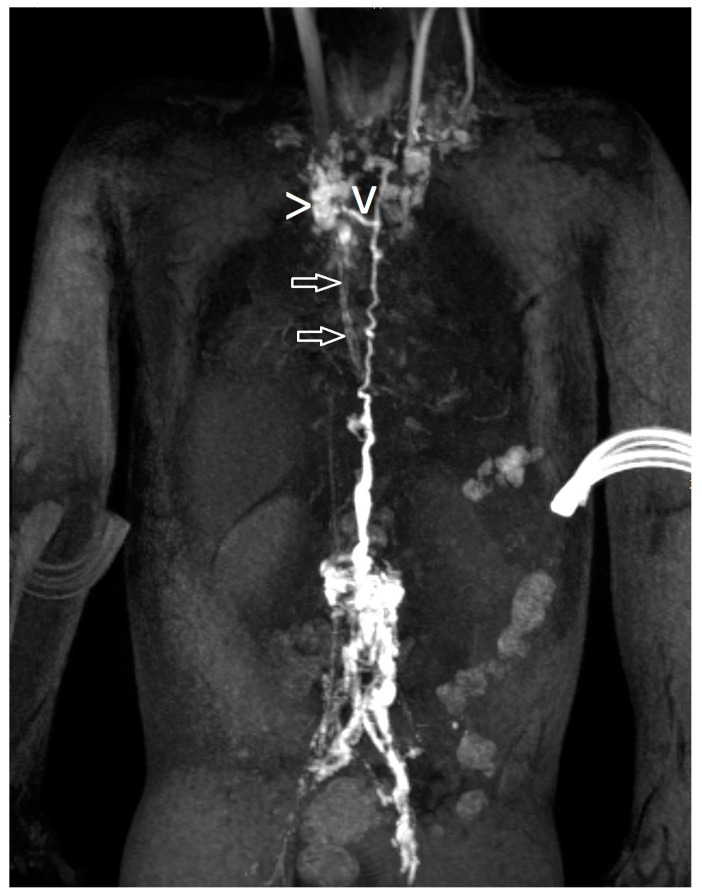
Lymphatic fistula (“>” points to the distinct structure, “v” is the fistula and the arrows show another lymphatic network).

**Figure 3 ijms-24-14827-f003:**
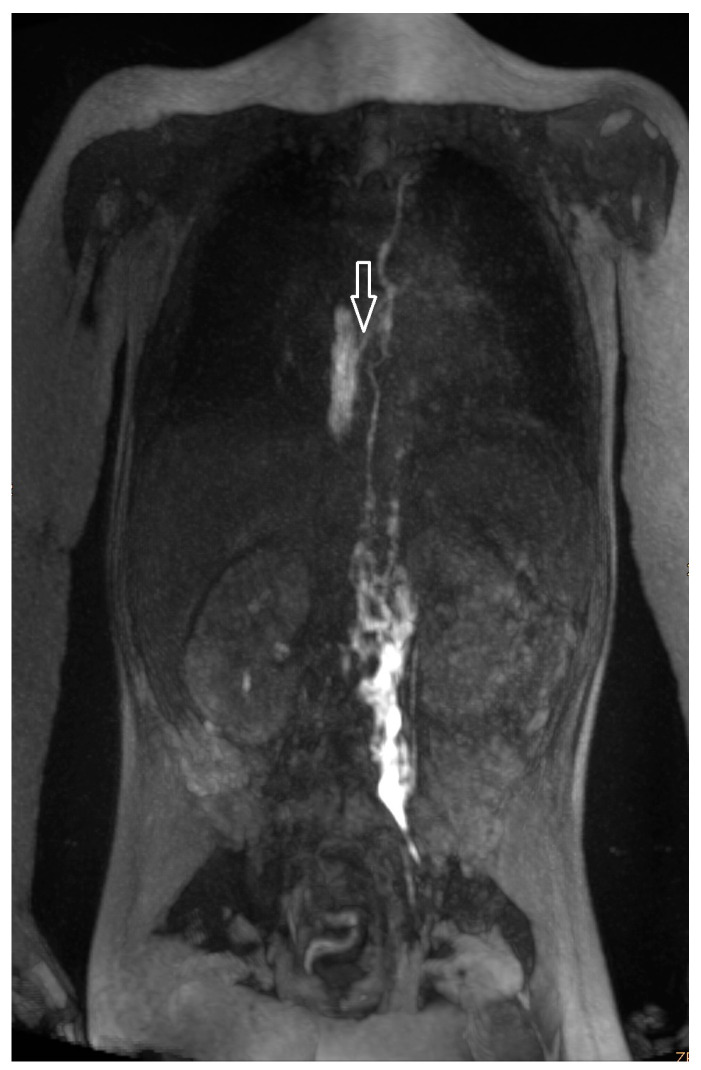
Lymphatic leak (marked with arrow).

**Figure 4 ijms-24-14827-f004:**
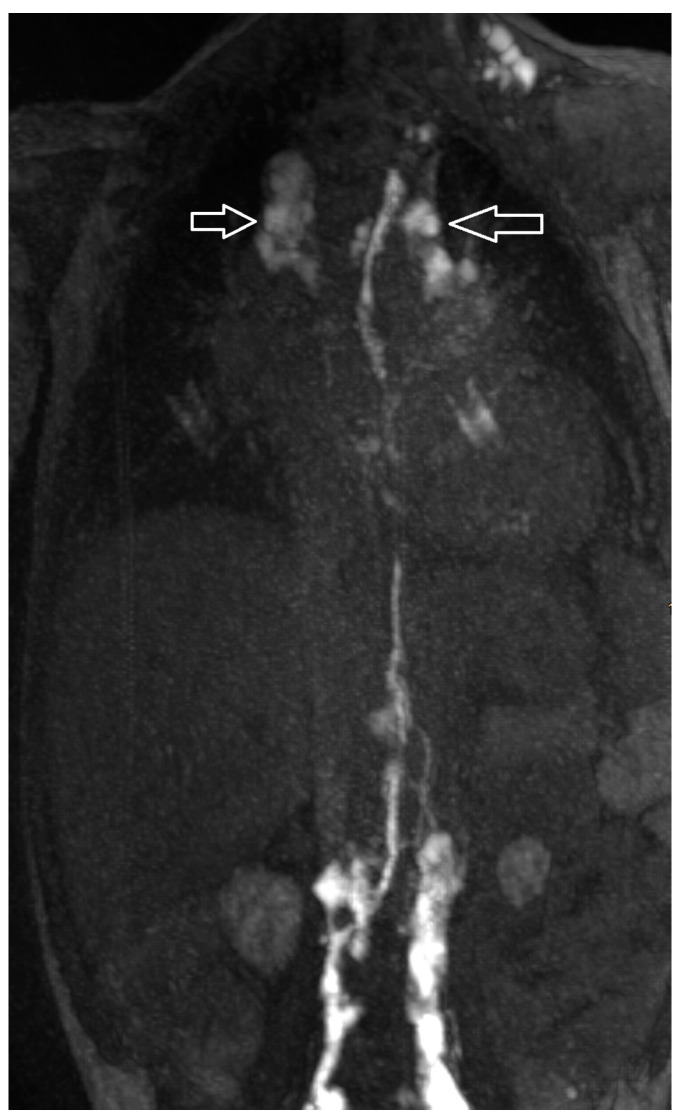
Pulmonary lymphatic perfusion syndrome (arrows point to abnormal lymphatic flow from the thoracic duct toward the lung parenchyma).

**Figure 5 ijms-24-14827-f005:**
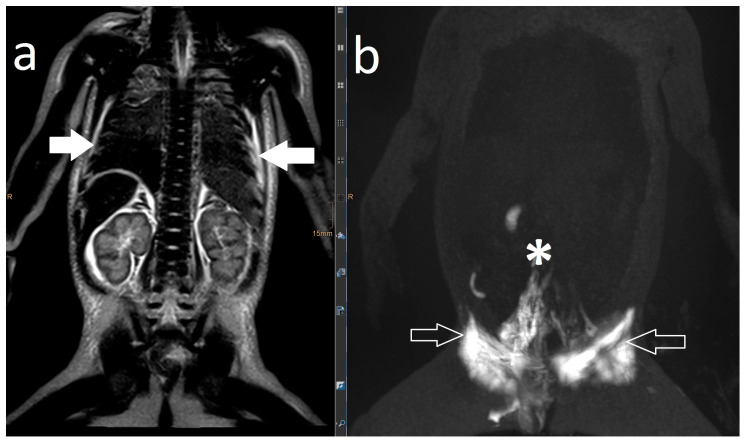
Central lymphatic flow disorders: abnormal central lymphatic flow, effusion in more than one compartment (**a**, arrows point to thoracic compartment) and the presence of dermal backflow (**b**, see arrows, * shows a stop in contrast movement).

**Figure 6 ijms-24-14827-f006:**
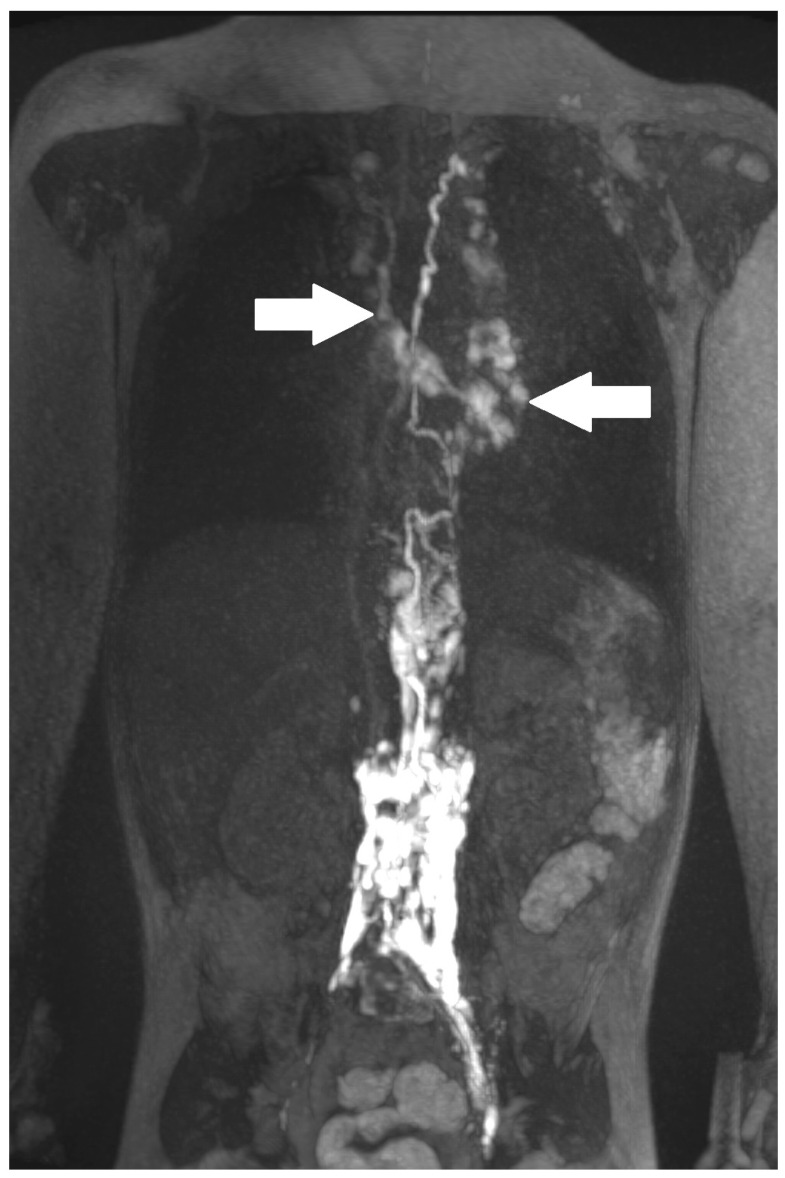
Typical findings of a patient with plastic bronchitis (arrows point to a leakage from the thoracic duct to the chest).

**Table 1 ijms-24-14827-t001:** Demographics and clinical history of study participants.

Patient Number	Gender *	Age **	Weight (kg)	Clinical Presentation	Congenital Heart Disease	Surgical Repair
Noonan syndrome:						
1	M	13 m	6.6	persistent congenital chylothorax with respiratory compromise	subvalvular-valvular-supravalvular PS	PV repair
2	F	11 m	7	pleural effusion with respiratory compromise	ASD II, double aortic arch, valvular PS	ASD closure, ligation of left aortic arch
3	F	6 a 10 m	12.4	persistent congenital chylothorax, respiratory compromise, ascites	ASD II, subvalvular- valvular- supravalvular PS, dysplastic aortic valve, RPA aLPA hypoplasia	RVOT-plastic, aortic valve repair
Chylothorax						
4	M	3 a 9 m	13.7		HLHS	Fontan ***
5	F	35 d	3.3		HLHR, LTGA	Norwood I/Sano shunt
6	M	22 d	3.4		HLHS, PAVPR	Norwood I/Sano shunt
7	M	2 a 9 m	13		TGA, SV, DILV	Fontan ***
8	F	3 a 3 m	13.9		unbalanced CAVC, TGA, TAPVR, right atrial isomerism	Fontan ***
9	F	44 d	3.7		TAPVR, Shone complex	TAPVR-Repair, end-to end-Anastomosis
10	F	24 d	3.4		hypopl. AA, subaortal stenosis, muscular VSD	AA reconstruction, modified Conno, VSD closure
11	M	3 a 9 m	12		Shone complex	combined Glenn + Fontan ***
12	F	9 m	5.8		Shone complex	Glenn
13	M	9 m	7.5		HLHS	Glenn
14	M	10 a 2 m	21.4		HLHS, Kabuki syndrome	Fontan ***
Protein-losing enteropathy						
15	F	5 a 11 m	10.7		Unb. CAVC, CoAc	Fontan ***
16	M	16 a 8 m	44		SV, TGA, hypopl. AA	Fontan *** (fenestration closed)
17	M	11 a 2 m	28.8		HLHS, LPA stenosis	Fontan *** (fenestration closed)
18	M	15 a 7 m	38.4		HLHS, recoarctation of aortae	Fontan *** (fenestration closed)
19	F	9 a 9 m	28.2		HLHS, PAPVR	extr. nonfenestrated Fontan
20	F	17 a 9 m	35.5		HLHS	Fontan *** (fenestration closed)
Plastic bronchitis						
21	M	3 a 8 m	14		HLHS	Fontan ***
22	F	6 a 11 m	21		HLHS	Fontan *** (fenestration closed)
23	M	7 a 6 m	20		HLHS	Fontan ***
24	M	8 a 1 m	21.5		HLHS	Fontan *** (fenestration closed)
25	F	9 a 5 m	26		HLHS	Fontan *** (fenestration closed)
26	F	12 a 9 m	25.8		unbalanced CAVC, HLHS	extr. nonfenestrated Fontan

Abbreviations: PS: pulmonary stenosis; ASD: atrial septal defect; RPA: right pulmonary artery; LPA: left pulmonary artery; HLHS: hypoplastic left heart syndrome; LTGA: levo-transposition of great arteries; PAVPR: partial anomalous pulmonary venous return; SV: single ventricle; DILV: double inlet left ventricle; CAVC: complete atrioventricular canal; TAPVR: total anomalous pulmonary venous return; hypopl. AA: hypoplastic aortic arch; VSD: ventricular septal defect; CoAc: coarctation of aortae; PHT: pulmonary hypertension; PV: pulmonary valve; RVOT: right ventricular outflow tract; * male (M), female (F); ** day (d), month (m), year (a); *** extracardiac fenestrated.

**Table 2 ijms-24-14827-t002:** Postoperative Chylothorax.

Pat. Nr.	MR Lymphography		Dynamic Contrast MR Lymphangiography			Preprocedural Management	Intervention/ Revision	Outcome (Time OP to LI/Time OP to Resolution/Time OP to Discharge)/Time Revision to Resolution
	LA-Type 1–4	T2-Findings (Abnormal Signal)	Lymphatic Flow	Etiology	Lymphatic Fistulas			
4	1	increased signal neck le	TD intact, reduced flow, retrograde flow towards pleural space le	T	no fistula visualized	MCT diet, octreotide	revision	resolution (57/73/82/8)
5	4	increased signal neck, axilla, mediastinum, hilum and body wall, pleural effusion le > ri, ascites	TD intact, retrograde flow towards hilum le, dermal backflow in abdominal wall, reduced central flow	CLFD	no fistula visualized	CT revision, furosemide, etacrynic acid, captopril,	MCT diet	resolution (21/44/63)
6	4	increased signal neck, axilla, mediastinum, hilum and body wall, pleural effusion, ascites	TD intact, delayed central lymphatic flow, retrograde flow towards pleural space ri > le, hilum le, dermal backflow in thoracic and abdominal wall	CLFD	no fistula visualized	CT revision	not indicated	died (20/-/-)
7	4	increased signal neck, mediastinum and hilum, bilateral pleural effusion	TD intact, retrograde flow towards mediastinum and lung parenchyma bilateral	PLPS	TD bilateral TV 2	CT revision, levosimedane	MCT diet, sandostatin, levosimedane	resolution (31/63/75))
8	3	increased signal neck, axilla le > ri, mediastinum le > ri, pleural effusion	TD intact, retrograde flow to mediastinum, lung ri	PLPS	no fistula visualized	MCT diet	revision	resolution (22/42/50/19)
9	4	increased signal neck, axilla, mediastinum, hilum and body wall, pleural effusion ri > le, ascites	TD discontinued from LV 3	CLFD	no fistula visualized	CT revision	MCT diet	resolution (24/44/59)
10	4	increased signal neck, mediastinum le > ri and hilum le > ri, pleural effusion le > ri, ascites	non-diagnostic	-	no fistula visualized	MCT diet	no intervention	resolution (16/29/33)
11	3	increased signal neck le > ri, mediastinum and hilum ri > le, pleural effusion	TD intact, retrograde flow towards mediastinum and hilum le	PLPS	no fistula visualized	diet modification	revision	resolution (42/62/71/20)
12	4	increased signal neck le > ri, mediastinum and hilum ri > le, pleural effusion ri > le, ascites	TD intact, retrograde flow towards mediastinum, hilum, lungs ri > le	PLPS	no fistula visualized	MCT diet, somatostatin	Glenn takedown	resolution (84/142/208/37)
13	4	increased signal neck, axilla, mediastinum, hilum and body wall, pleural effusion	retrograde flow towards hilum le, dermal backflow into thoracic wall	CLFD	TV 7–10 to hilum/ mediastinum	MCT diet	not performed yet	recurrent CT
14	1	increased signal neck le.	non diagnostic	-	no fistula visualized	LPA-stent placement	not indicated	resolution of CT

Abbreviations: CT: chylothorax; LA: lymphatic abnormality; TD: thoracic duct; T: traumatic; CLFD: central lymphatic flow disorder; PLPS: pulmonary lymphatic perfusion syndrome; MCT: medium chain triglyceride; LI: lymphatic imaging; LPA: left pulmonary artery; OP: operation; ri: right; le: left; LV: lumbar vertebra; TV: thoracic vertebra; Pat. Nr.: patient number.

**Table 3 ijms-24-14827-t003:** Plastic bronchitis.

Pat. Nr.	MR Lymphography		Dynamic Contrast MR Lymphangiography	Preprocedural Management	Intervention		Outcome *
	LA-Type 1–4	T2-Findings (Abnormal Signal)	Lymphatic Flow	Lymphatic Fistulas			
21	4	increased signal mediastinum, hilum, lung le > ri	retrograde flow from TD with diffuse mediastinal and peribronchial perfusion	thoracic vertebrae 4–10 to left mediastinum	MCT diet, macitentan, alteplase inhalation, salbutamol, diuretics	glue embolization of TD	cast free after intervention (FU 1 a 4 m) tapering off sildenafil
22	4	increased signal (mediastinum), hilum/lung ri	retrograde lymphatic flow to mediastinum and lung parenchyma ri	from hilum to ri lung	fat reduced diet, sildenafil, alteplase/ fluticasone inhalation	selective glue embolization of 2 branches of TD	cast free after intervention (FU 4 a), cessation of fat-reduced diet,
23	4	increased signal mediastinum, hilum, lung ri > le	retrograde lymphatic flow towards lung parenchyma & mediastinum ri > le and peribronchial perfusion	fistula visualized	fat reduced diet, sildenafil, diuretics, alteplase inhalation, azithromycine	selective glue embolization of fistulas	cast free after intervention (FU 1 a 4 m)
24	4	increased signal at the mediastinum and hilum bilateral	retrograde lymphatic flow towards mediastinum and hilum bilaterally	no fistula visualized	diuretics salbutamol inhalation. Budesonide	TD decompression	cast free after OP, albumin normalized (FU 3 a 5 m)
25	4	increased signal neck, axilla, mediastinum, hilum, lung ri, fistula from liver to ri lung	little contrast	no fistula visualized	lisinopril, diuretics, spironolactone, bisoprolol, sildenafil, diuretics, alteplase inhalation.	not performed yet	still casts
26	4	increased signal neck, axilla, mediastinum, hilum, lung	retrograde lymphatic flow towards mediastinum, hilum and lung parenchyma with peribronchial perfusion ri	from hilum to ri lung	spironolactone, sildenafil, diuretics, alteplase inhalation, hydrochlorothiazide	not performed yet	still casts

Abbreviations: TD thoracic duct; LA: lymphatic abnormality; OP: operation; MCT: medium-chain triglyceride; FU: follow up; * day (d), month (m), year (a); ri: right; le: left; Pat. Nr.: patient number.

**Table 4 ijms-24-14827-t004:** Protein-losing enteropathy.

Pat. Nr.	MR Lymphography		Dynamic Contrast MR Lymphangiography		Preprocedural Management	Intervention	Outcome *
	LA-Type 1–4	T2-Findings (Abnormal Signal)	Lymphatic Flow	Lymphatic Fistulas			
15	1	increased signal neck le	TD intact, no retrograde flow	no	budesonide, sildenafil, macitentan, diuretics	no due to comorbidity	no improvement
16	1	increased signal neck le, hilar lymphadenopathy	TD intact, no retrograde flow	no	budesonide, macitentan, diuretics enalapril	glue embolization	albumin improved (FU 2 a 7 m)
17	3	increased signal neck and hilum	TD intact, no retrograde flow	no	budesonide, sildenafil, losartan	TD decompression	scheduled for transplantation
18	1	increased signal neck le	TD intact, no retrograde flow	no	budesonide, sildenafil, spironolactone	no (patient disapproval)	no improvement
19	1	increased signal neck le	TD intact, no retrograde flow	no	enalapril	no	transient improvement
20	1	slightly increased signal neck le	TD intact, no retrograde flow	no	enalapril, aldactone, budesonide	no	no improvement

Abbreviations: TD thoracic duct; LA: lymphatic abnormality; FU: follow up; * day (d), month (m), year (a); ri: right; le: left; Pat. Nr.: patient number.

**Table 5 ijms-24-14827-t005:** Noonan syndrome. Imaging findings as well as management and outcome of children with Noonan syndrome.

Pat. Nr.	MRL		DCMRL		Preprocedural Management	Intervention	Outcome *
	LA-Type 1–4	T2-Findings (Abnormal Signal)	Lymphatic Flow	Lymphatic Fistulas			
1	4	increased signal neck, mediastinum, hilum, perihilar, interstitial lung parenchyma, pleural effusion	abnormal, perfusion to the lung, intercostal flow, dilated lymphatic networks in neck, mediastinum, hilum and perihilar	in the right apical lung	diuretics, sildenafil, non-invasive ventilatory support	lymphatic intervention	Resolution of CT (FU: 10 m, cessation of ventilatory support.
2	2	increased signal neck	abnormal, prevertebral, lymphovenous collaterals	no	diuretics, sildenafil, milrinone, levosimedan, invasive ventilatory support.	no indication, ASD closure	resolution of pleural effusion
3	4	increased signal neck, mediastinum, hilum, interstitial lung parenchyma, body wall edema, pleural effusion, ascites	abnormal, perfusion to the lung, intercostal flow, dermal backflow,	TV 6–9 bilateral into the lung	diuretics, sildenafil, MCT diet, non-invasive ventilatory support	no	persistent chylothorax, respiratory support

Abbreviations: TD thoracic duct; TV: thoracic vertebra; CT: chylothorax; ASD: atrial septal defect; LA: lymphatic abnormality; MCT: medium-chain triglyceride; FU: follow up; * day (d), month (m), year (a); Pat. Nr.: patient number.

## Data Availability

All data generated or analyzed during this study are included in this published article.

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
