# Peer review of "Non-Contrast MR Lymphography and Intranodal Dynamic Contrast MR Lymphangiography in Children with Congenital Heart Disease—Imaging Findings as well as Impact on Patient Management and Outcome"

_ijms, 2023, doi:10.3390/ijms241914827_

Round 1
Reviewer 1 Report
The authors present a retrospective review of non-contrast MR lymphangiography and intranodal dynamic contrast MR-lymphangiography in children with congenital heart disease. The lymphatic system is a major player in the maintenance of cardiovascular health and the current study highlights the importance of MR lymphangiography on guiding treatment modalities and patient outcomes. The work is well written, and the results are well described and discussed. The one thing I would want to see, if the authors have the information, is how the lymphatic function and structure improved/changed after the intervention(s) and how that correlated with patient outcomes. However, this being a retrospective study, it is understandable if the authors do not have that data. In that case, a short discussion will be beneficial as well.
Author Response
We do not do lymphatic imaging after an intervention routinely.
So unfortunately there is no data on lymphatic function and structure after an intervention.
Reviewer 2 Report
1. The authors present an in-depth study on the use of MR-lymphography and dynamic contrast MR-lymphangiography. However, the explanation of the techniques and their complexities remains quite surface level. A more detailed description of the technicalities involved in these procedures would add significant value to the study.
2. While the authors mention the learning curve associated with these procedures, there is no clear discussion on how this learning curve was managed and how it affected the patients' outcomes in the early stages. This lack of information presents a critical gap in the study.
3. The authors only report one minor complication, which gives an unrealistic picture of the risks involved in these procedures. The manuscript could be strengthened with a broader discussion on potential complications and how they were mitigated.
4. The results highlight the differential imaging findings between patients with different conditions, but the explanation of these differences is vague. The authors fail to offer a robust interpretation or clinical implications of these findings.
5. The study seems to emphasize the need for additional imaging techniques in cases where intranodal dynamic contrast MR-lymphangiography may not be sufficient. However, they do not offer much evidence to support this need. This suggestion could use more concrete backing.
6. The section on patient outcomes lacks clarity. The connection between the interventions and the reported outcomes is not well established, and the role of spontaneous resolution versus surgical intervention is not defined.
7. The study’s conclusions, while they may hold significance, rely on a small, limited patient sample. Therefore, generalizability is questionable. Additionally, the manuscript fails to present long-term follow-up data to support the efficacy and safety of the techniques discussed.
8. The absence of a control group or comparison with other imaging techniques weakens the robustness of the study. The inclusion of such would offer a more comprehensive perspective.
The manuscript's clarity could be enhanced by simplifying sentence structures and ensuring consistent terminology. Consistent writing conventions, such as number usage, should be adhered to. The presence of technical jargon and grammatical errors necessitates more thorough proofreading to make the paper more accessible and readable.
Round 2
Reviewer 2 Report
The authors addressed most of my concerns, I have no more question.